# Enhanced Anti-Babesia Efficacy of Buparvaquone and Imidocarb When Combined with ELQ-316 In Vitro Culture of *Babesia bigemina*

**DOI:** 10.3390/ph18020218

**Published:** 2025-02-06

**Authors:** Natalia M. Cardillo, Nicolas F. Villarino, Paul A. Lacy, Joseph S. Doggett, Michael K. Riscoe, Carlos E. Suarez, Massaro W. Ueti, Chungwon J. Chung

**Affiliations:** 1Animal Disease Research Unit, USDA-ARS, 3003 ADBF, WSU, Pullman, WA 99163, USA; lacyp@wsu.edu (P.A.L.); massaro.ueti@usda.gov (M.W.U.); 2Department of Veterinary Microbiology and Pathology, Washington State University, Pullman, WA 99164, USA; 3Program in Individualized Medicine, Department of Veterinary Clinical Sciences, College of Veterinary Medicine, Washington State University, Pullman, WA 99164, USA; nicolas.villarino@wsu.edu; 4VA Portland Healthcare System, 3710 SW US Veterans Hospital Road, Portland, OR 97239, USA; doggettj@ohsu.edu (J.S.D.); riscoem@ohsu.edu (M.K.R.); 5School of Medicine, Division of Infectious Diseases, Oregon Health & Science University, 3181 SW Sam Jackson Park Road, Portland, OR 97239, USA; 6Department of Microbiology and Molecular Immunology, Oregon Health & Science University, 3181 SW Sam Jackson Park Road, Portland, OR 97239, USA

**Keywords:** in vitro culture, efficacy, therapies, apicomplexan, potency

## Abstract

**Background/Objectives:** *B. bigemina* is a highly pathogenic and widely distributed tick-borne disease parasite responsible for bovine babesiosis. The development of effective and safe therapies is urgently needed for global disease control. The aim of this study is to compare the effects of endochin-like quinolone (ELQ-316), buparvaquone (BPQ), imidocarb (ID), and the combinations of ID + ELQ-316 and BPQ + ELQ-316, on in vitro survival of *B. bigemina.* **Methods:** Parasites at a starting parasitemia level of 2%, were incubated with each single drug and a combination of drugs, ranging from 25 to 1200 nM of concentration over four consecutive days. The inhibitory concentrations, 50% (IC50%) and 99% (IC99%), were estimated. Parasitemia levels were evaluated daily using microscopic examination. Data were statistically compared using the non-parametrical Kruskall–Wallis test. **Results:** All drugs tested significantly inhibited (*p* < 0.05) the growth of *B. bigemina* at 2% parasitemia. The combination of ID + ELQ-316 exhibited a lower mean (IC50%: 9.2; confidence interval 95%: 8.7–9.9) than ID (IC50%: 61.5; confidence interval 95%: 59.54–63.46), ELQ-316 (IC50%: 48.10; confidence interval 95%: 42.76–58.83), BPQ (IC50%: 44.66; confidence interval 95%: 43.56–45.81), and BPQ + ELQ-316 (IC50%: 27.59; confidence interval: N/A). Parasites were no longer viable in cultures treated with the BPQ + ELQ-316 combination, as well as with BPQ alone at a concentration of 1200 nM, on days 2 and 3 of treatment, respectively. **Conclusions:** BPQ and ID increase the babesiacidal effect of ELQ-316. The efficacy of these combinations deserves to be evaluated in vivo, which could lead to a promising and safer treatment option for *B. bigemina*.

## 1. Introduction

*Babesia bovis* and *Babesia bigemina* are the primary causative agents of bovine babesiosis, an economically impactful disease affecting cattle worldwide [1,2,3,4]. While *B. bovis* has been extensively studied due to its association with cerebral injury, high mortality rates, and significant economic losses, *B. bigemina* is typically linked to milder acute hemolytic disease, but it can still be highly pathogenic. It is widely distributed and transmitted transovarically by a broad range of *Rhipicephalus* ticks [5].

Calves exhibit a natural resistance to *B. bigemina*; however, the disease can be acute in older or immunocompromised animals. Infection with *B. bigemina* typically manifests more benignly, but severe consequences can arise due to hemolytic anemia, with mortality rates reaching up to 50% without treatment. Additionally, after acute signs resolve, cattle can survive as carriers, suffering from recurrent infections or dying from secondary complications, ultimately impacting animal production [6].

Treatment of *B. bigemina* with imidocarb (ID) can effectively cure the infection, but it leaves the animal susceptible to reinfection. For this reason, reduced drug dosage levels have sometimes been recommended [7]. However, under field conditions, the use of drugs at subtherapeutic low concentrations, whether prophylactically or in chemoimmunization programs, increases the risk of resistance emergence [8,9,10]. Additionally, imidocarb is associated with residue issues and is not widely available, including in the USA and Europe [7].

Buparvaquone (BPQ) has shown promise in treating *Babesia bovis* and *B. bigemina*, as well as other apicomplexan parasites [11,12,13,14,15,16,17,18,19,20,21]. BPQ selectively inhibits the Qo quinone-binding site of the parasite’s mitochondrial cytochrome b electron transport system, leading to its lethal effect [16,22,23,24].

ELQ-316 is another new drug that proved effective in in vitro treatment of *Theileria* spp. and *Babesia* spp. parasites. ELQ-316 is an endochin-like quinolone compound (ELQ). ELQ-316 selectively inhibits the Qi quinone-binding site of the parasite’s mitochondrial cytochrome bc_1_ complex electron transport system and has shown high efficacy against *Plasmodium falciparum*, *Babesia microti,* and *Toxoplasma gondii*. Different ELQ compounds demonstrated a strong antimalarial effect both in vitro and in vivo, along with parasite selectivity, chemical and metabolic stability, desirable pharmacokinetics, and low toxicity to mammalian cells [25].

Effective treatment of babesiosis is crucial for maintaining the health and productivity of livestock. The lack of approved drugs in many countries for controlling babesiosis, and the emergence of drug resistance in *Babesia* species underscores the need for new treatments. BPQ and ELQ-316 may enhance treatment options, providing effective and safe alternatives, as previously demonstrated [26].

It is well established that there are differences in the susceptibility to antiprotozoal drugs between larger *Babesia* species (e.g., *B. bigemina*) and smaller species (such as *B. bovis*) [1,2]. This indicates that the effects of drugs may not be the same for *B. bovis* and *B. bigemina*, and therefore, drugs should be tested independently of data obtained previously from *B. bovis*. The aim of the present study is to evaluate the efficacy of ID, BPQ, ELQ-316, as well as the combinations of BPQ + ELQ-316 and ID + ELQ-316, in inhibiting the replication of *B. bigemina* in vitro.

## 2. Results

### 2.1. Comparative Inhibitory Effects of BPQ, ID and ELQ-316 and the Combinations ELQ-316 + BPQ and ELQ-316 + ID Against B. bigemina In Vitro Replication

The median and rate survival kinetics of in vitro *B. bigemina* cultures over 96 h of incubation in the presence of various concentrations (nM) of BPQ, ID, ELQ-316, and the combinations ELQ-316 + BPQ and ELQ-316 + ID, starting at 2% PPE, are shown in Table 1 and Figure 1. At 25 nM, the survival rate observed was comparable for all drugs and drug combinations. At 75 nM, a significant difference was only observed between ID and both drug combinations (BPQ + ELQ-316 and ID + ELQ-316). At 150 nM, significant differences were observed between both ELQ-316 combinations and the three drugs tested individually. A similar pattern was observed at 300 nM, except for ELQ-316, which did not differ (*p* > 0.05) from the other drugs and combinations tested. At 600 and 1200 nM, all drugs effectively killed *B. bigemina*.

### 2.2. Drug Potency

The mean and 95% confidence intervals of IC50% and IC99% values were calculated to compare the potency of the drugs on the growth of in vitro *B. bigemina* cultures. Results are presented in Table 2. The combination of ELQ-316 + ID demonstrated the lowest IC50%, indicating higher potency in killing *B. bigemina*. However, it required seven times the IC50% dose (IC99%/IC50%) to achieve 0% survival. The combination of BPQ + ELQ-316 exhibited the second greatest potency. Surprisingly, the concentration required to achieve 0% survival was very close to the IC50%. The potencies of BPQ and ELQ-316 were similar, and the concentration of drugs required to kill completely was between 3 and 3.5 times the IC50%. ID demonstrated the lowest potency, which was similar to that of its combination with ELQ-316; it also required 7 times the IC50% to eliminate the parasites completely.

### 2.3. Time and Drug Concentration Required to Reach 0% Survival After Treatment

The survival of *Babesia bigemina* in in vitro cultures after drug treatments of varying durations and doses for each drug and combination is presented in Table 3.

Parasites treated with BPQ at 1200 nM became non-viable after 48 h, while those treated at doses ranging from 300 nM to 1200 nM showed no survival after 72 h. Cultures treated with drug concentrations above 75 nM maintained this trend, reaching 0% survival after 96 h. A similar pattern was observed in cultures treated with ELQ-316.

The combination of BPQ + ELQ-316 demonstrated a superior effect, achieving 0% survival more rapidly. No parasites were detected after 48 h of treatment with BPQ + ELQ-316 at 1200 nM, after 72 h at doses ranging from 300 nM to 1200 nM, and after 96 h at doses ranging from 75 nM to 1200 nM.

In cultures treated with ID + ELQ-316, no parasites were detected after 72 h at concentrations of 600 nM and 1200 nM, and after 96 h at concentrations ranging from 150 nM to 1200 nM. Additionally, after 24 h of incubation in media without ID + ELQ-316, cultures previously treated with concentrations ranging from 75 nM to 1200 nM showed no surviving parasites.

## 3. Discussion

New, safe, and effective chemotherapies are urgently needed for the global control of bovine babesiosis. Combination drug strategies have proven significantly more effective in eliminating parasites and, importantly, reducing the risk of developing drug resistance compared to single-drug therapies [27].

In this study, we compared the drugs BPQ and ELQ-316, which appear to exert synergistic effects by targeting two distinct sites (Qo and Qi, respectively) in the parasite’s mitochondrial cytochrome *bc*_1_ complex. Both combinations tested (ELQ-316 + BPQ and ELQ-316 + ID) demonstrated nearly 100% efficacy in killing *B. bigemina* at 150 nM over 96 h of treatment. Similar results were observed for ELQ-316 and BPQ monotherapies at the same concentration. Notably, the combination of BPQ + ELQ-316 was the fastest to achieve 0% survival, reaching this endpoint at 96 h with 75 nM and at 48 h with 1200 nM.

The IC50 values for the combinations of BPQ + ELQ-316, BPQ and ELQ-316 were similar. Our calculated IC50 values for ELQ-316 were significantly higher than those reported for *B. bigemina* and other related apicomplexans in previous studies [28]. For instance, ELQ-316 demonstrated IC50 values of 7.97 nM for *Besnoitia besnoiti* and 0.66 and 0.35 nM for *Toxoplasma gondii* tachyzoites [25,29]

Significant differences were observed between BPQ and ID, consistent with findings from previous studies on *B. bovis* reported by Cardillo, Lacy, et al. (2024) at 1% PPE and 2% PPE [12,30]. Importantly, a significant difference was also observed between the IC50 values of BPQ + ELQ-316, ELQ-316, and BPQ, with the BPQ + ELQ-316 combination demonstrating the highest potency against *B. bovis* [30].

Regarding the time to achieve 0% survival, the combination of ELQ-316 + BPQ was significantly faster and more effective at lower concentrations compared to the ELQ-316 + ID combination. The ELQ-316 + BPQ combination reached 0% survival at 48 h with 1200 nM and at 72 h with concentrations as low as 300 nM. In contrast, the ELQ-316 + ID combination required 600 nM to achieve 0% survival within the same 72 h timeframe.

In this study, we found that *B. bigemina* (IC50%: 27.25 nM) was more sensitive to BPQ than *B. bovis* (IC50: 50.01 nM and 77.06 nM), as reported in our previous work with cultures started at 1% and 2% PPE, respectively [12,30]. This finding contrasts with those of Nugrara et al. (2019), who reported significantly higher drug potency for both parasites, indicating that *B. bovis* was much more sensitive than *B. bigemina* (IC50: 135 ± 41 nM and 488 ± 30 nM, respectively). Nugrara’s study utilized starting cultures at 1% PPE and measured parasitemia after 96 h of treatment with a fluorescence-based method using SBYR Green 1 stain [20].

Conversely, a significant difference in parasite survival was observed with the combination of ELQ-316 + ID at 25 nM compared to the other drugs (*p* < 0.05), which may explain the low IC50 of 9.25 nM (95% CI: 8.67–9.89) for this combination. This represents a substantial improvement in potency compared to each drug individually (ID IC50: 61.49 nM [95% CI: 59.54–63.46]; ELQ-316 IC50: 48.10 nM [95% CI: 42.76–58.83]), suggesting a notable synergistic effect when both drugs are used together. Interestingly, the low IC50 obtained with the combination of ELQ-316 + ID is comparable to that reported by Rizk et al. (2023) for the combination of Diaminazene aceturate (DA) + ID against *B. bigemina* [31]. The authors demonstrated a synergistic effect of both drugs, with a combination index value (CI) of 0.76, indicating synergy (CI < 0.9) based on the Chou–Talalay method (Chou, 2006). It can be speculated that both drugs may possess complementary mechanisms of action to kill or inhibit the growth of Babesia, a possibility that may also apply to the combination of ELQ-316 + ID.

In this study, ID required a higher concentration to achieve near 0% survival within the same evaluation period (96 h) than the other drugs, with effective results observed at concentrations above 300 nM. While *B. bigemina* remains more sensitive to ID than *B. bovis* [12], a significant improvement in drug potency was observed with the combination of ELQ-316. Although the mechanism of action of ID is not well understood [32], as suggested by Silva et al. (2020), the potential presence of a resistant subpopulation may have significantly influenced survival rates when combined with ELQ-316, thereby impacting overall results. Interestingly, the doses and concentrations of each drug required to achieve 0% survival (IC99%) were similar, in proportion, to those observed by Cardillo et al. (2024b) against *B. bovis*. Among these, ID combined with ELQ-316 and ID alone required the highest doses—almost seven times the IC50—to completely eliminate the parasites in vitro in our study. In the study by Cardillo et al. (2024b), nearly four times the IC50% was required to achieve the same results. This suggests that ELQ-316 may have an additive effect when used with ID, facilitating faster elimination of susceptible parasites. However, a more resistant population that requires higher doses to achieve 0% survival, especially ID, may exist.

Importantly, since ELQ-316 is safer than ID, it could provide a promising option for reducing the therapeutic doses of ID required, thereby minimizing toxic side effects in animals and decreasing residues retained in edible tissues. This combination approach enhances efficacy and aligns with the growing need for safer and more effective treatments in livestock management.

Similar to our previous report about *B. bovis* [12], a residual effect against parasites was observed after treatment with all drugs and combinations tested. This suggests a persistent effect, likely related to the extent of the impairment or metabolic changes, which depend on the concentration of each drug used. We observed a dose and time-dependence survival effect which align with previous studies on *B. bovis* using endochin-like quinolones, tulathromycin drugs [28,33], BPQ [12], and artemisinin [34]. Silva et al. (2020) reported that *B. bovis*, *B. caballi*, and *T. equi* were unable to grow in in vitro cultures in the presence of ELQ-316, regardless of their initial percentage of parasitemia (*p* < 0.05). However, they observed that adding another endochin-like quinolone (ELQ-300) to *B. bigemina* cultures at an initial parasitemia of 2% did not lead to a rapid decrease in parasitemia, in contrast to the results seen when the initial parasitemia was set at 0.2%. The same effect was observed with ELQ-316 against *B. caballi*. The authors speculated that this could be due to the presence of pre-existing resistant parasite subpopulations of *B. bigemina* and *B. caballi*, capable of surviving the initial drug-inhibitory treatment with ELQ-300 and ELQ-316. It was recently shown that genetic alterations in the Q_i_ binding site of the cytochrome *bc*_1_ complex (Cytb) of *B. microti* is associated with resistance to ELQ-316, suggesting that this cytochrome gene may be as a potential target for the ELQ drugs [35]. This finding supports the idea that combining chemotherapeutics can offer significant advantages, including enhanced effectiveness, reduction in dosage (which may lead to fewer toxic side effects), and lower likelihood of resistance or/and recrudescence. Drugs that produce synergistic effects and inhibit distinct pathways, can also help to prevent anti-drug resistance development [26].

The therapeutic efficacy of in vitro drug screening assays against *Babesia* spp. is influenced by various factors, including species, strain, and size of the screening parasites, the culture conditions (such as the medium used, percentage of parasitemia [PPE], hematocrit, and the presence or absence of serum), and the methods used to calculate PPE [36,37]. These variables can significantly impact the calculated IC50 values of the tested drugs across different studies [38,39]. There is a critical need for consensus on standardized international methods to improve comparability and reproducibility of results. This would facilitate more accurate comparisons of drug efficacy and enhance our understanding of therapeutic potential across different *Babesia* species and treatments.

Overall, the ability of all drugs tested to inhibit *B. bigemina* growth in time and dose concentration was superior to the previous study on *B. bovis*. At concentrations of 600 nM and above, all drugs were 100% effective in killing *B. bigemina*. These findings reinforce the notion that large *Babesia* species (like *B. bigemina*) are more susceptible to antiprotozoal drugs than smaller species (such as *B. bovis*). However, no single treatment guarantees a radical cure [1,2]. Differences in size and metabolism between small and large *Babesia* spp., may lead to variations in drug target and biochemical mode of action [40].

This study focused on the asexual cycle within the definitive host, and therefore, we were unable to directly observe the effects of the treatment on the stages that occur in the tick. Our experimental design was limited to the asexual stages within the erythrocytes, so we cannot determine whether the drug affects other stages outside the host. As for the parasite staging, all observed stages were trophozoites of the Texas strain, which have been maintained in long-term laboratory cultures.

## 4. Materials and Methods

### 4.1. Chemical Reagents

Buparvaquone (98% pure) was obtained from Combi-Blocks, Inc. (San Diego, CA, USA). ID (VETRANAL™, Supelco^®^ Buchs, Switzerland) was used as a positive control for the in vitro inhibition assays of *B. bigemina*, using the same protocol described for BPQ below. The purity of ID was determined to be >98% by proton nuclear magnetic resonance spectroscopy and high-performance liquid chromatography (HPLC), as indicated in the certificate of analysis.

ELQ-316 was synthesized using methods previously described by Nilsen et al. (2014) and identified by proton nuclear magnetic resonance (^1^H NMR), with a purity of ≥95% confirmed by reversed-phase high-performance liquid chromatography (RP-HPLC) [41].

BPQ, ELQ-316, and ID were diluted in 100% dimethyl sulfoxide (DMSO) to prepare stock solutions, which were stored at room temperature until use. Working solutions were freshly prepared in a parasite culture medium on each test day, just before being added to the parasite cultures.

### 4.2. Parasites Culture

*Babesia bigemina* (Puerto Rico Strain: Goff et al. 1998) were grown in long-term microaerophilic stationary-phase cultures at 37 °C in an atmospheric condition of 5% CO_2_, 5% O_2_, and 90% N_2_, as previously described [42]. *B. bigemina* were cultured in 96 well-culture plates, in 190 μL per well of PL culture media (100 mL = pH 7.2; 29 mL F-12K Nutrient Mixture + L-glutamine (Life Technologies, Carlsbad, California, USA), 29 mL Stable Cell IMDM (Sigma Aldrich, St. Louis, MO, USA), 2 mL 0.5 M TAPSO (Sigma Aldrich, St. Louis, MO, USA, 0.5 mL Antibiotic Antimycotic solution 100× (Sigma Aldrich, St. Louis, MO, USA), 1 mM calcium chloride (Sigma Aldrich, St. Louis, MO, USA), 100 μL Antioxidant Supplement 1000× Sigma Aldrich, 1 mL Insulin-Transferrin-Selenium-Ethanolamine 100× (Sigma Aldrich, St. Louis, MO, USA), 1 mL 50% Glucose (Teknova, Hollister, CA, USA), 500 μL L-glutamine 200 mM (GIBCO, Grand Island, NY, USA), and 40% bovine serum. The cultures were maintained in 5% cells volume of erythrocytes.

### 4.3. In Vitro Growth of Initial Inhibitory Assay

The in vitro inhibitory efficacy of BPQ, ELQ-316, and ID was evaluated against the growth of *B. bigemina*, after starting at a percentage of parasitized erythrocytes (PPE) of 2%. The parasitemia level was selected based on previously validated protocols to ensure the presence of a sufficient number of parasites for meaningful drug efficacy testing, while also avoiding excessively high parasitemia that could affect the consistency of the results. The chosen range is between 0.5% and 2%. *B. bigemina* were cultured in media containing various final concentrations (25, 75, 150, 300, 600, and 1200 nM) of BPQ, ELQ-316, ID, and the combinations of BPQ + ELQ-316 and ID + ELQ-316, diluted in 100% DMSO. Cultures with DMSO (1 μL) and without drug compounds served as positive controls for parasite growth, while additional wells containing uninfected bovine RBCs were used as negative controls.

Parasite cultures were replenished daily with fresh culture media (190 μL/well) containing the respective drug concentrations. The experiments were conducted in triplicate for each concentration and control over 96 h (4 days). PPE was monitored daily by counting parasites in Hema 3 Stat Pack (Fisher Scientific, Pittsburgh, PA, USA) stained thin blood smears (GBS). Before the daily media change, 190 μL of supernatant was removed from each well, and the remaining RBC layer at the bottom was suspended. A 1 μL sample was taken from each well to prepare a thin smear, and the number of infected red blood cells was counted by visually examining 5000 erythrocytes on each slide. Morphological characteristics were also observed. Drug responsiveness of the parasites was measured as the percentage of parasitemia after every 24 h exposure to drug concentrations, up to the 96 h mark.

### 4.4. Viability After Treatment

At 96 h after the initial treatment, fresh media without drugs were added to all culture wells, along with 5 μL of fresh RBCs. This drug-free medium was replaced daily for the next five days to assess whether the cultures remained viable and could continue growing without the drug. Both quantitative and qualitative parasitemia levels were determined through a microscopic examination of Hema 3 Stat Pack (Fisher Scientific) stained thin blood smears.

### 4.5. Statistical Analysis

Values of parasitemia were counted daily, and the percentage of survival was calculated, and media comparisons between all tested drugs were performed using the Mann–Whitney test or Kruskall–Wallis test.

The doses of a drug that produced 50% inhibition (IC50%) relative to the control population and the maximal inhibitory concentration (IC99%) were determined for BPQ, ELQ-316, ID, and the combinations, at drug concentrations ranging from 25 to 1200 nM, in 24 h, 48 h, 72 h, and 96 h post-incubation. Total inhibitory concentrations (IC99%) were determined as the drug doses required to reduce parasite growth to the same level observed in non-infected erythrocytes (approximately 0.1%). The survival curves were compared using a log-rank (Mantel–Cox) test. The statistical significance was set at <0.05. GraphPad Prism ver. 7 software for Windows (Graphpad Software Inc., San Diego, CA, USA) was used for the statistical analysis.

## 5. Conclusions

The findings reinforce the efficacy of the tested drugs in inhibiting *B. bigemina* parasites in vitro and highlight the promising potential of drug combinations as alternative control strategies. While in vitro efficacy can serve as a reliable predictor of therapeutic potential in vivo, drug screening assays often lack host-related variables tied to pharmacokinetic and pharmacodynamic factors, as well as interactions at non-target sites [43]. Therefore, the therapeutic activity of these potent candidates must be validated in target animal species, bovines.

Variations in parasite species, strains, or culture conditions during in vitro drug screening, along with differing methods for counting parasitemia, can explain the diverse pharmacological interactions observed with drug combinations in various studies. Further research is needed to clarify these differences and deepen our understanding of drug efficacy, the interplay between drug mechanisms of action, and the potential mechanisms of parasite resistance.

The combination therapy involving ELQ-316 with ID or BPQ significantly reduced parasitemia of *B. bigemina* more effectively than the full-dose monotherapies. Therefore, future studies are needed to investigate the mechanisms by which these drugs interact to inhibit *Babesia* growth. While the potential role of BPQ in human babesiosis, as well as the use of ELQ-316 alone or in combination with BPQ, is beyond the scope of the current study, it is worth noting that these treatments may warrant further investigation in the context of human infections. This represents an important avenue for future research, especially considering the increase of its global relevance.

This study offers a potential novel approach to address the toxicity and resistance associated with high ID doses. However, further research is required to determine the clearance of this combination from the products of treated animals.

## Figures and Tables

**Figure 1 pharmaceuticals-18-00218-f001:**
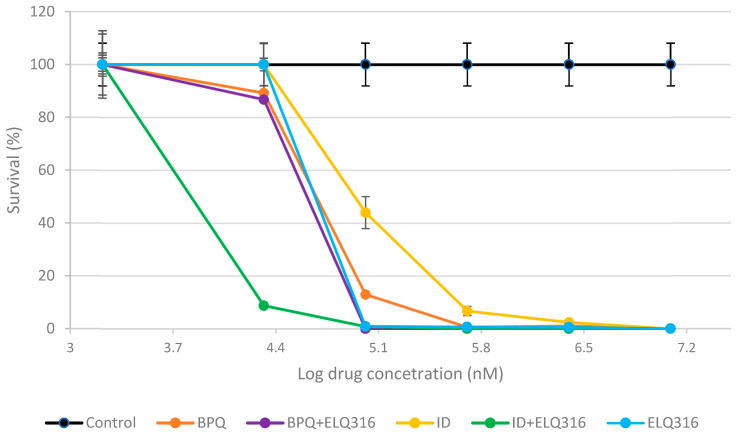
Comparative survival kinetics (median %) of in vitro *B. bigemina* cultures during 96 h of incubation with different concentrations (nM) of ELQ-316, BPQ, the BPQ + ELQ-316 combination, the ID + ELQ-316 combination, and ID, starting at a 2% parasitemia (PPE), are presented. Error bars represent the standard deviation (n = 3 experiments) for each drug and combinations tested. X axis points represent 0 nM, 25 nM, 75 nM, 150 nM, 300 nM, 600 nM, and 1200 nM in logarithmic scale.

**Table 1 pharmaceuticals-18-00218-t001:** Comparative survival rate (%) of *B. bigemina* cultures after 96 h in vitro incubation, starting at 2% PPE, in the presence of different concentrations of BPQ, ID and ELQ-316, and their combinations.

Drug Treatment (nM)	BPQ	ID	ELQ-316	ID + ELQ-316	BPQ + ELQ-316
	Median Range (%)				
25	89.26 (84.30–92.98)	100 (88–100)	100 (89.26–100)	8.80 (6.94–11.9)	86.78 (79.34–100)
75	13.02 (8.68–13.64) ^a,b^	44.01 (38.43–49.59) ^b^	0.87 (0.62–0.99) ^a,b^	0.87 (0.62–0.99) ^a,c^	0 ^c^
150	0.62 (0.37–0.74) ^a^	6.76 (2.48–11.03) ^a^	0.65 (0.5–1.12) ^a^	0 ^c^	0 ^c^
300	1.12 (0.99–1.86) ^a^	2.48 (1.24–3.72) ^a^	0.53 (0.37–0.87) ^a,b^	0 ^c^	0 ^c^

^a^, ^b^, ^c^ Drugs with different letters differed (*p* < 0.05).

**Table 2 pharmaceuticals-18-00218-t002:** In vitro drug potencies (IC50% and IC99%) for *B. bigemina* at 96 h of incubation.

Drugs	IC50% (nM)		IC99% (nM)	
	Mean	95% CI	Mean	95% CI
ID + ELQ-316	9.248	(8.667–9.887)	65.52	(61.35–69.81)
BPQ + ELQ-316	27.59	N/A	34.23	N/A
BPQ	44.66	(43.56–45.81)	156.4	(148.9–164.4)
ELQ-316	48.10	(42.76–58.83)	133	(102.2–155.6)
ID	61.49	(59.54–63.46)	441.4	(392.3–498)

N/A, not applicable: CI were not reported for this concentration’s groups because the mathematical models did not fit the data adequately.

**Table 3 pharmaceuticals-18-00218-t003:** Drug concentrations ranges and times at which 0% survival was attained.

Single Drugs and Combination Treatments	Time Post-Treatment with Drug (h)			Time Post-Treatment Without Drug (h)				
	48	72	96	24	48	72	96	120
BPQ	-	1200	600 to 1200	(--------------- 150 to 1200 ------------)				
BPQ + ELQ-316	1200	300 to 1200	(------------------------- 75 to 1200 ----------------------)					
ELQ-316	-	1200	600 to 1200	(--------------- 150 to 1200 ------------)				
ID + ELQ-316	-	600 to 1200	(--------- 150 to 1200 -------)		(------ 75 to 1200 -----)			
ID	-	-	600–1200	300–1200	(----- 150 to 1200 ----)			

## Data Availability

The raw data supporting the conclusion of this article will be made available by the authors, without undue reservation.

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
