# Peer review of "Enhanced Anti-Babesia Efficacy of Buparvaquone and Imidocarb When Combined with ELQ-316 In Vitro Culture of Babesia bigemina"

_pharmaceuticals, 2025, doi:10.3390/ph18020218_

Round 1
Reviewer 1 Report
Comments and Suggestions for Authors
In the current study "ELQ-316 enhances the efficacy of buparvaquone and imidocarb in controlling the in vitro growth of Babesia bigemina" authors use a variety of drugs as monotherapy and combined therapy to evaluate the effect of the drugs in in vitro culture over 96h. While the study design and overall results and their presentation is satisfactory, there are some concerns which must be addressed ti make the paper suitable for publication.
1. The title of the paper is misleading as it suggests that ELQ makes BPQ as well as ID more (lines 115-120)
2. The survival % of the combination drugs should be absoloute and clearly stated in Table 1- it is unclear why the superscipt "c" is used when there should be a number to the survival % in combinatorial drug regimen
3. A positive control drug should have been used in the experiments to ensure that the experiment has been performed properly.
4. It is unclear why the drug treatment was done on parasites at 2% - is there a reason why this number was chosen? what was the staging of the parasite when treated - were they synchronized? clarify -
5. A few lines must be devoted to the lifecycle, in vitro culture stages and parasite progression of the parasite. It remains unclear that the drug is targeting all stages of the parasite in culture or is it a specific stage.
Author Response
For research article
Response to Reviewer 1 Comments |
Point-by-point response to Comments and Suggestions for Authors |
Comments 1: In the current study "ELQ-316 enhances the efficacy of buparvaquone and imidocarb in controlling the in vitro growth of Babesia bigemina" authors use a variety of drugs as monotherapy and combined therapy to evaluate the effect of the drugs in in vitro culture over 96h. While the study design and overall results and their presentation is satisfactory, there are some concerns which must be addressed to make the paper suitable for publication. Comments 1. The title of the paper is misleading as it suggests that ELQ makes BPQ as well as ID more (lines 115-120) |
Response 1: Thank you for your feedback. I understand your concern regarding the wording of the title. The intent was to convey that ELQ-316 enhances the efficacy of both Buparvaquone (BPQ) and Imidocarb (ID) specifically in controlling the in vitro growth of Babesia bigemina To address this, we changed the title as follows: ‘Enhanced anti-babesia efficacy of buparvaquone and imidocarb when combined with ELQ-316 in vitro culture of Babesia bigemina.
|
Comment 2: The survival % of the combination drugs should be absolute and clearly stated in Table 1- it is unclear why the superscipt "c" is used when there should be a number to the survival % in combinatorial drug regimen. |
Response 2: Thank you for your comment. Regarding the combination drugs, we can confirm that no parasites survived, resulting in a 0% survival rate. The superscript "c" indicates a statistically significant difference in comparison to the other drug tests. If another drug test (either alone or in combination) shares at least one letter in common, there is no significant difference between them.
Comments 3: A positive control drug should have been used in the experiments to ensure that the experiment has been performed properly. Response 3: Thank you for your valuable suggestion regarding the use of a positive control drug in our experiments. We agree that incorporating a positive control is an important aspect of experimental design, as it helps validate the effectiveness of the treatment under investigation. In our study, we focused on evaluating the specific effects of ELQ-316, Buparvaquone, and Imidocarb against Babesia bigemina in vitro. The positive control in our study is the Buparvaquone and Imidocarb treatments alone, based on previously established data for Buparvaquone and specially Imidocarb, which are known to be effective in treating Babesia infections. Imidocarb is the chemotherapy drug approved for B. bigemina control. We also ensured that our experimental setup followed rigorous protocols to minimize potential errors.
Comments 4: It is unclear why the drug treatment was done on parasites at 2% - is there a reason why this number was chosen? what was the staging of the parasite when treated - were they synchronized? clarify - Response 4: Thank you for your thoughtful comments. We understand the need for clarification regarding the choice of 2% parasitemia for drug treatment, as well as the staging of the parasites during treatment. The 2% parasitemia level was selected based on previously validated and established protocols for other Babesia in vitro cultures. The level which is typically used to ensure that enough parasites are present for meaningful drug efficacy testing, while also avoiding overly high parasitemia that might affect the consistency of the results is between 0.5 to 2. Low levels may affect the growing, and higher level may affect the viability of the culture. This parasitemia level has been shown to be effective in our previous studies and allowed us to observe the effects of treatment under controlled conditions. Regarding the staging of the parasites, they are all trophozoites of Texas strain, maintained in long term laboratory cultures. We synchronized the parasitemia level in the cultures to start, making a dilution that assure the same parasitemia level in all test wells.
Comments 5: A few lines must be devoted to the lifecycle, in vitro culture stages and parasite progression of the parasite. It remains unclear that the drug is targeting all stages of the parasite in culture or is it a specific stage. Response 5: Thank you for your valuable comments. Regarding the concern about the parasite stages, we would like to clarify that in the definitive host (such as cattle or other vertebrate hosts), only the trophozoite stage of Babesia is observed within the erythrocytes. The other stages, such as merozoites, schizonts, and the sexual stages (gametocytes), occur in the tick, which serves as the vector. Since our study focused on the asexual cycle in the definitive host, it was not possible to observe directly the effects of the treatment on the stages that occur in the tick. Our experimental design was limited to the asexual stages within the erythrocytes, so we cannot determine if the drug affects other stages outside the host. |
Reviewer 2 Report
Comments and Suggestions for Authors
LINE 54: "....REDUCED DRUG DOSAGE LEVELS...." (BETTER THAN DRUG 'DOSES' LEVELS)
NICE IN VITRO STUDY
NICE REVIEW AND ATTEMPTS TO 'RECONCILE' CONFLICTING RESULTS OF DIFFERENT STUDIES.
NICE HANDLING IN BOTH DISCUSSION SECTION AND CONCLUSIONS OF THE PROBLEMATIC NATURE OF BABESIAE INFECTIONS AND NEED FOR CONTINUED RESEARCH AND SEARCH FOR/DEVELOPMENT OF IMPROVED METHODS OF TREATMENT (WITH GOAL OF CURE).
THE REVIEWER WONDERS IF THERE IS A ROLE FOR BUPARAQUONE IN HUMAN BABESIOSIS CASES. DITTO USE OF ELQ-316 ALONE OR IN COMBINATION WITH BUPARVAQUONE IN HUMAN CASES. PERHAPS OUT OF SCOPE OF CURRENT AUTHORS WORK...BUT MIGHT BE WORTH A 'MENTION'
Author Response
Thank you very much for your kind and thoughtful feedback. We made changes to the manuscript.
We greatly appreciate your positive comments on the study, particularly regarding the handling of conflicting results and the discussion of the challenges associated with Babesia infections. We also acknowledge your recognition of the importance of continued research and the development of improved treatment methods.
Regarding your suggestion about the potential role of Buparvaquone in human babesiosis cases, as well as the use of ELQ-316, either alone or in combination with Buparvaquone, we agree that these are important considerations. While these topics are indeed outside the scope of our current study, we appreciate your insight and will be sure to mention the potential relevance of these treatments for human babesiosis in the discussion section of the paper. It is certainly an area that warrants further exploration in future research.
Once again, thank you for your valuable suggestions and constructive feedback. We incorporated your recommendation into the manuscript to ensure a more comprehensive perspective in lines 340 to 344.
Reviewer 3 Report
Comments and Suggestions for Authors
Dear Authors
The manuscript entitled ‘ELQ-316 enhances the efficacy of buparvaquone and imidocarb in controlling the in vitro growth of Babesia bigemina’ investigated the effects of ELQ-316, buparvaquone and imidocarb and their combinations on B. bigemina in vitro assays. It was determined that all drugs used in the study were effective against the parasite at different rates. In addition, ELQ-316+buparvaquone and ELQ-316+imidocarb combinations were found to be more effective against B. bigemina due to their synergetic effect. Considering that B. bigemina is a vector-borne pathogen and causes significant economic losses in the world due to clinical manifestations in cattle, the results obtained in this manuscript are considered promising. Suggestions regarding the manuscript are presented below. In addition, it is suggested that the authors revise and correct the references section according to “Instructions for Authors” of the journal (https://www.mdpi.com/journal/pharmaceuticals/instructions#references) .
Abstract
In the abstract section of the manuscript, information about the purpose of the study and the results obtained is given. This section is written in a way that the reader can easily understand.
Introduction
The researchers gave brief information about the symptoms seen in hosts in infections caused by Babesia bigemina and vector tick species. They also gave information about the drugs used in the treatment of B. bigemina and their effect mechanisms. In addition to these, the researchers also gave information about the aim of the study. It is considered that the information given in the introduction of the manuscript is sufficient. However, it was observed that there were no implementations on B. bovis in the manuscript and it was not understood why information about this parasite was given. Other suggestions about this section are given below.
Line. 41 Please do not start the paragraph with an abbreviation.
Lines 84-86. This paragraph is considered inappropriate for the introduction. It is suggested to transfer this section to the conclusion or discussion section of the manuscript.
Results
The authors have presented the data obtained in detail in the results section of the study. Different typefaces were used in Table 2 and this part should be corrected.
Discussion
In the discussion section of the manuscript, the authors discuss the information obtained within the aim of the study by comparing it with the results of different studies. It is considered that the information presented in this section is sufficient. Suggestions related to this section are given below.
Line 146. Please add a comma before ‘and’.
Materials and methods
In the material and method section of the manuscript, the researchers gave information about all the applications made within the scope of the study. It is considered that the information given in this section is sufficient. Suggestions related to this section are given below.
Line 256 Please do not start the paragraph with an abbreviation.
Line 270 “CO2, O2, and N2” should be written as “CO2, O2, and N2”
References
It was observed that the parasite names in the references used in the study were not italicized. It is recommended that the references should be carefully reviewed from beginning to end and necessary corrections should be made.
Kind regards.
Author Response
Response to Reviewer 3 Comments |
Point-by-point response to Comments and Suggestions for Authors |
Dear Authors The manuscript entitled ‘ELQ-316 enhances the efficacy of buparvaquone and imidocarb in controlling the in vitro growth of Babesia bigemina’ investigated the effects of ELQ-316, buparvaquone and imidocarb and their combinations on B. bigemina in vitro assays. It was determined that all drugs used in the study were effective against the parasite at different rates. In addition, ELQ-316+buparvaquone and ELQ-316+imidocarb combinations were found to be more effective against B. bigemina due to their synergetic effect. Considering that B. bigemina is a vector-borne pathogen and causes significant economic losses in the world due to clinical manifestations in cattle, the results obtained in this manuscript are considered promising. Suggestions regarding the manuscript are presented below. In addition, it is suggested that the authors revise and correct the references section according to “Instructions for Authors” of the journal (https://www.mdpi.com/journal/pharmaceuticals/instructions#references). Thank you for your positive and thoughtful feedback on our manuscript entitled "ELQ-316 enhances the efficacy of buparvaquone and imidocarb in controlling the in vitro growth of Babesia bigemina." We are grateful for your recognition of the significance of our findings, particularly the synergistic effects of the ELQ-316+buparvaquone and ELQ-316+imidocarb combinations in enhancing the efficacy against B. bigemina. We also appreciate your mention of the economic importance of B. bigemina as a vector-borne pathogen and the potential impact of our findings on controlling its spread in cattle. Regarding your suggestion to revise and correct the references section according to the journal’s "Instructions for Authors," we carefully reviewed and made the necessary adjustments to ensure full compliance with the guidelines. Abstract In the abstract section of the manuscript, information about the purpose of the study and the results obtained is given. This section is written in a way that the reader can easily understand. Introduction The researchers gave brief information about the symptoms seen in hosts in infections caused by Babesia bigemina and vector tick species. They also gave information about the drugs used in the treatment of B. bigemina and their effect mechanisms. In addition to these, the researchers also gave information about the aim of the study. It is considered that the information given in the introduction of the manuscript is sufficient. However, it was observed that there were no implementations on B. bovis in the manuscript and it was not understood why information about this parasite was given. Other suggestions about this section are given below. Response: Thank you for your valuable feedback. The inclusion of Babesia bovis in the manuscript was made in reference to two previous studies that we performed on B. bovis, evaluating the same drugs (Buparvaquone, ELQ-316 and Imidocarb) and demonstrated their efficacy against B. bovis. Given the close relationship between B. bovis and B. bigemina, we included this information to highlight that the same treatments might also be effective against B. bigemina, as supported by these earlier findings. Line. 41 Please do not start the paragraph with an abbreviation. Response: We appreciate your attention to detail regarding the use of abbreviations. We made the changes in the manuscript. Lines 84-86. This paragraph is considered inappropriate for the introduction. It is suggested to transfer this section to the conclusion or discussion section of the manuscript. Response: Thank you for your helpful feedback. We understand your concern regarding the paragraph on lines 84-86 and appreciate your suggestion to move it to the conclusion or discussion section. We initially included this paragraph in the introduction because it is common in some papers, particularly those focusing on human studies, to briefly mention the results or implications early in the introduction to provide context for the research. However, we recognized that this may not align with the typical structure of an introduction and we removed from the section. Results The authors have presented the data obtained in detail in the results section of the study. Different typefaces were used in Table 2 and this part should be corrected. Response: Thank you for your constructive feedback. We appreciate your acknowledgment of the detailed presentation of the data in the results section. Regarding the formatting issue in Table 2, we corrected the use of different typefaces to ensure consistency throughout the manuscript. Discussion In the discussion section of the manuscript, the authors discuss the information obtained within the aim of the study by comparing it with the results of different studies. It is considered that the information presented in this section is sufficient. Suggestions related to this section are given below. Line 146. Please add a comma before ‘and’. Response: Thank you for your attention to detail. We added the comma before "and" in line 146, as per your suggestion. Materials and methods In the material and method section of the manuscript, the researchers gave information about all the applications made within the scope of the study. It is considered that the information given in this section is sufficient. Suggestions related to this section are given below. Line 256 Please do not start the paragraph with an abbreviation. Response: Thank you for your valuable suggestion. We have made the necessary changes as recommended. Line 270 “CO2, O2, and N2” should be written as “CO2, O2, and N2” Response: Thank you for your valuable suggestion. We have made the necessary changes as recommended. References It was observed that the parasite names in the references used in the study were not italicized. It is recommended that the references should be carefully reviewed from beginning to end and necessary corrections should be made. Response: Thank you for your careful review of the manuscript. We appreciate your observation regarding the formatting of parasite names in the references. We have thoroughly reviewed the references and made the necessary corrections to ensure that all parasite names are italicized, as per the journal's formatting guidelines. Thank you again for your attention to detail. Kind regards. |
Round 2
Reviewer 1 Report
Comments and Suggestions for Authors
Thank you for addressing the queries. I am satisfied with the responses - but I would prefer that for readers to understand the context of your study, include the responses in your main text - esp for lifecycle (and the limitation that this stdy has only investigated asexual stages) under discussion and about why 2% parasitemia has been chosen should be included under methods
Author Response
Dear Reviewer, Thank you so much for your revision, changes have been made in the new version of the manuscript attached.
Kind regards
